# Using cfDNA and ctDNA as Oncologic Markers: A Path to Clinical Validation

**DOI:** 10.3390/ijms241713219

**Published:** 2023-08-25

**Authors:** Jonathan Dao, Patrick J. Conway, Baskaran Subramani, Devi Meyyappan, Sammy Russell, Daruka Mahadevan

**Affiliations:** 1Long School of Medicine, University of Texas Health San Antonio, San Antonio, TX 78229, USA; 2Mays Cancer Center, University of Texas Health, San Antonio, TX 78229, USA; 3Graduate School of Biomedical Sciences, University of Texas Health San Antonio, San Antonio, TX 78229, USA

**Keywords:** liquid biopsy, cfDNA, ctDNA

## Abstract

The detection of circulating tumor DNA (ctDNA) in liquid biopsy samples as an oncological marker is being used in clinical trials at every step of clinical management. As ctDNA-based liquid biopsy kits are developed and used in clinics, companies work towards increased convenience, accuracy, and cost over solid biopsies and other oncological markers. The technology used to differentiate ctDNA and cell-free DNA (cfDNA) continues to improve with new tests and methodologies being able to detect down to mutant allele frequencies of 0.001% or 1/100,000 copies. Recognizing this development in technology, the FDA has recently given pre-market approval and breakthrough device designations to multiple companies. The purpose of this review is to look at the utility of measuring total cfDNA, techniques used to differentiate ctDNA from cfDNA, and the utility of different ctDNA-based liquid biopsy kits using relevant articles from PubMed, clinicaltrials.gov, FDA approvals, and company newsletters. Measuring total cfDNA could be a cost-effective, viable prognostic marker, but various factors do not favor it as a monitoring tool during chemotherapy. While there may be a place in the clinic for measuring total cfDNA in the future, the lack of standardization means that it is difficult to move forward with large-scale clinical validation studies currently. While the detection of ctDNA has promising standardized liquid biopsy kits from various companies with large clinical trials ongoing, their applications in screening and minimal residual disease can suffer from lower sensitivity. However, researchers are working towards solutions to these issues with innovations in technology, multi-omics, and sampling. With great promise, further research is needed before liquid biopsies can be recommended for everyday clinical management.

## 1. Introduction

Screening and early diagnosis are essential strategies to decrease mortality, treatment cost, and disease burden of cancer [1]. Diagnosis of various cancers at earlier stages increases five-year survival and cure rate, and diagnosis at later stages has dramatically higher medical costs [2,3]. After diagnosis, determining prognosis and molecular profiling plays a central role in patient management, especially as precision medicine becomes the status quo of treatment selection [4,5]. 

As described by the U.S. Preventive Services Task Force (USPSTF), active screening methods, such as mammograms for breast cancer, pap smears for cervical cancer, or colonoscopy for colorectal cancer, have been an effective means of early detection in older patients [6]. Mammograms prevent 21 deaths per 10,000 women screened over 10 years, pap smears prevent up to 8.34 deaths per 1000 women screened to the age of 65, and colonoscopies prevent up to 28 deaths per 1000 adults screened from the age of 45 to 65 [7,8,9]. Additionally, low-dose computed tomography is recommended as a screening method for lung cancer in patients aged 50–80 years who have a 20 pack-year smoking history and currently smoke or have quit within the past 15 years which would prevent 503 deaths per 100,000 adults over a lifetime of screening. Even with these screening methods, limited sensitivity, specificity, and low incidence of cancer keeps them from being recommended to healthy individuals, and each method only screens for one cancer type [6]. 

For diagnosing, grading, and profiling tumors, solid tissue biopsy is the gold standard. However, solid biopsies are skill-intensive, invasive, risk missing tumor heterogeneity, and have low sensitivity [10]. Further, due to its invasive nature, a solid biopsy cannot be carried out for screening and/or monitoring during treatment. For cancers without screening, diagnosis with solid biopsy comes at later stages when patients are symptomatic. This contributes to the poor prognosis of lung, pancreatic, ovarian cancers, and other cancers [11,12,13]. 

Liquid biopsy (LB) is a technique that is being studied at every step of cancer management for screening, minimal residual disease (MRD), monitoring for recurrence in the adjuvant setting, and treatment selection for advanced cancer. LB mostly uses peripheral blood, but can also include other body fluids, such as urine, saliva, feces, pleural effusions, ascites, and cerebrospinal fluid (CSF) [14]. LB methods have been used to detect circulating tumor DNA (ctDNA), RNA (ctRNA), circulating tumor cells (CTCs), microRNA (miRNA), extracellular vesicles (EVs), and tumor-educated platelets (TEPs). 

As a method of early detection, LB can be used to screen for multiple cancers at once, becoming more cost-effective than the array of screening methods used currently [15]. For use in diagnosis, LB has several advantages over solid tissue biopsy. LBs are non-invasive, samples are easier to obtain, and result in faster turnaround time [16]. In addition, LB can be performed repeatedly to monitor for response and progression and achieve better detail of the tumor’s spatial heterogeneity [10,17]. However, LB does not allow for histological evaluation, which would require a solid biopsy, and most panels have a limited number of genes. While there are some FDA-approved LB tests for prognosis and treatment selection, such as CellSearch, cobas *EGFR* Mutation Test v2, Guardant360 CDx, and FoundationOne Liquid CDx [18], LB tests for screening and diagnosis are still being clinically validated [4,17]. 

Here, we review methods of isolating ctDNA, as well as their recent applications for the purposes of screening, diagnosing, and management through ctDNA kinetics of various tumor types.

## 2. Circulating Free DNA

Circulating cell-free DNA (cfDNA) are extracellular fragments of dsDNA between 120–220 bp long, centered around 167 bp, which is associated with the nucleosome pattern of cfDNA in apoptosis [19]. cfDNA has a short half-life that varies from 4 min to 2 h, which lends itself to applications in monitoring. cfDNA can be found in various body fluids, such as blood, urine, or cerebrospinal fluid [20,21,22,23]. Under normal conditions, cfDNA can come from apoptosis, neutrophil extracellular traps (NETs), and erythroblast enucleation [24,25,26]. In plasma, cfDNA originates from granulocytes (32%), erythrocyte progenitors (30%), lymphocytes (12%), monocytes (11%), vascular endothelial cells (9%), and hepatocytes (1%) [27]. The cfDNA can be increased in normal physiological processes, such as physical exercise, or in pathological processes that increase cell death, such as inflammation, sepsis, or myocardial infarction [28,29,30,31]. 

The ctDNA is the fraction of cfDNA that originates from tumor cells, which comes from three sources: apoptosis, necrosis, and active secretion. While ctDNA can come from apoptosis with fragment lengths similar to healthy patients, ctDNA is more fragmented or shorter than cfDNA [20,32,33]. In necrosis, chromatin does not fragment in a nucleosome pattern but is cleaved at random generating fragments of various sizes, which contributes to both shorter and longer fragments of >10,000 bp [34]. While apoptosis and necrosis are straightforward answers to the source of ctDNA, the concentration of cfDNA cannot be fully explained by the death of tumor cells [35]. However, the exact mechanisms of the actively secreted DNA are not fully understood with possible mechanisms being exosomes or amphisomes [36,37]. Overall, in patients with cancer, clinically relevant information from cfDNA could include a concentration of cfDNA/ctDNA, methylation patterns, and gene mutations (Figure 1).

## 3. Total cfDNA Concentration

In healthy individuals, the concentration of cfDNA in plasma is between 0–10 ng/mL with the serum concentration of cfDNA being 10 times higher [38,39,40]. Most cfDNA in serum concentration comes from the process of clotting in the collection tube, which makes plasma better for clinical applications [41,42]. In patients with cancer, the concentration of cfDNA in plasma can be from 0 to over 1000 ng/mL [39,40,43,44]. This varies amongst cancer types. For instance, patients with gliomas had less ctDNA than other solid tumors, such as the pancreas, colon, breast, or ovary [45]. The concentration of cfDNA also varies with stage, where metastatic cancers have more cfDNA followed by locally advanced and then localized cancers [40]. A major challenge in the implementation of cfDNA in clinical use is the variance from the lack of standardization in methodology, such as collecting samples from serum versus plasma, extracting cfDNA with different kits, measuring samples using different techniques, time of collection, and timing for response [36,46,47].

Total cfDNA concentration has been studied extensively for its application as an early detection biomarker. For instance, a meta-analysis showed that using cfDNA concentration for diagnosis of lung cancer yielded a pooled sensitivity of 80% and specificity of 77% [36]. A study on differentiating prostate cancer from benign hyperplasia showed a sensitivity of 73.2% and specificity of 72.7% using cfDNA concentration, which aligns with a more recent study stating the low sensitivity and specificity result in poor diagnostic utility [37,48]. A study on breast cancer screening found that there was a significant difference in total cfDNA concentration between invasive breast cancer vs. disease-free subjects, but no significant difference between in situ cancer, benign tumors, and invasive cancer [49]. A study on gastrointestinal (GI) tract malignancies, including esophageal cancer, stomach cancer, and colorectal cancer, found an overall sensitivity of 75.8% and specificity of 95.8% [50]. The sensitivity and specificity of total cfDNA concentration in most studies is equal to that of serum proteins, which are generally not recommended for use in diagnosis or screening. Increased cfDNA in numerous processes and lower levels of cfDNA in earlier stages of cancer also decrease the utility of cfDNA concentration for these applications. 

However, total cfDNA concentration could be an effective measure for prognosis determination and patient monitoring. Tumor markers (e.g., PSA, CEA, CA-125, etc.) are used in prognosis determination or monitoring during/after treatment for patients with cancer. In pancreatic cancer, the American Society of Clinical Oncologists (ASCO) guidelines recommend that CA19-9 in serum and CT are used for monitoring during treatment [51]. However, CA19-9 is not recommended for prognostic prediction of outcomes, and 5–10% of the population are Lewis antigen negative, with little to no secretion of CA19-9 [52,53]. In prostate cancer, PSA continues to be a mainstay in screening, prognosis, and monitoring after treatment in conjunction with other modalities [54,55]. However, the use of PSA may be associated with overdiagnosis and overtreatment for indolent cancers [56]. In breast cancer, ASCO guidelines state that CA15-3 and CA27-29 are serum proteins with prognostic value. CA15-3, CA27-29, and carcinoembryonic antigen (CEA) could be used for monitoring patients during active therapy in conjunction with other modalities. There were no serum proteins that were recommended for detecting recurrence after therapy [57]. In colorectal cancer (CRC), CEA is a recommended marker for determining prognosis, monitoring during treatment, and monitoring postoperatively for evaluation of metastatic disease [58].

Total cfDNA concentration has been shown in studies of pancreatic, prostate, breast, and colorectal cancer to be a cost-effective tool for prognosis, monitoring during, and monitoring for recurrence, which could fill the gaps in utility that some serum proteins have. There are also studies on lung cancer, which do not have any recognized biochemical markers (Table 1).

In a study of 74 patients with advanced/metastatic pancreatic cancer, high levels of total cfDNA were associated with new distant metastasis (NDM) with 91% sensitivity and 95% specificity. Researchers also found that total cfDNA concentration was associated with worse progression-free survival (PFS) and overall survival (OS). In two cases, they found that cfDNA was elevated in the first month after treatment of chemotherapy while CA19-9 levels were below detection prior to detection of NDM on CT scans [59]. In a meta-analysis subgroup of 3 studies, elevated levels of pre-operative total cfDNA had prognostic value associated with recurrence after surgery, and post-operative elevation of total cfDNA was also associated with recurrence and poor prognosis [60]. 

For prostate cancer, in a meta-analysis of 23 studies, total cfDNA concentration was found to be a poor diagnostic tool as previously discussed, but total cfDNA had similar prognostic value to PSA for PFS and OS [61]. Interestingly, PSA and cfDNA were independent of each other, and the combination of the two increased the discrimination between indolent vs. lethal prostate cancer [61]. 

A study of 194 patients with metastatic breast cancer (MBC) comparing total cfDNA, CTCs, and CA15-3 showed that cfDNA was a good predictor of OS and the best predictor of PFS [62]. For monitoring response to treatment, total cfDNA concentration had a discriminatory power that was 10% higher than CA15-3. However, the overall accuracy was low for all modalities studied [62]. A study of 117 patients with MBC, found that elevated cfDNA concentration was independently associated with unfavorable PFS and OS after adjusting for CTCs. They also found that decreased cfDNA compared to baseline levels was associated with increased treatment response [63]. In two studies of locally advanced breast cancer, there were differing conclusions regarding the concentration of cfDNA where one study found that increased cfDNA after chemotherapy was associated with improved treatment response [68]. Similarly, the other study found increased cfDNA after one cycle, but over the course of 6–8 cycles decreased cfDNA was associated with tumor response [69]. 

In a meta-analysis of seven studies on metastatic colorectal cancer (mCRC), high cfDNA levels were associated with poor overall survival [64]. In a study of 43 patients with mCRC, researchers also found that elevated cfDNA prior to treatment was correlated with poor overall survival and found that patients with worse response to treatment had significantly higher cfDNA concentrations at the end of treatment than those with stable disease [65]. In various gastrointestinal malignancies, elevated pre- and postoperative cfDNA concentrations were associated with tumor recurrence. In patients with tumor progression after treatment, cfDNA levels were more sensitive than CEA in monitoring for recurrence, with elevated cfDNA being detected earlier than elevated serum CEA levels [50].

In non-small cell lung cancer (NSCLC), a meta-analysis of 22 studies found that patients with elevated cfDNA concentration tend to have shorter PFS and OS [66]. A study within that meta-analysis of 218 patients found that individuals in the top third of cfDNA concentration before treatment showed a significantly shorter PFS and OS than patients in the lower two-thirds. However, cfDNA concentration during or after treatment did not have any association with the response to treatment [70]. Another study of 177 patients found that cfDNA had prognostic value predicting PFS and OS. Again, they found that there was no association of cfDNA concentration during or after treatment with the response to treatment [67]. In contrast, a number of other studies have found significant differences in cfDNA concentration in different response groups [38,71,72,73] (Table 1).

Overall, total cfDNA concentration has promise as a biomarker for prognosis in cancers discussed above, often being as accurate or better than serum protein biomarkers currently used for pancreatic, prostate, and breast cancer [59,61,62]. The meta-analysis for CRC noted that due to a lack of CEA measurements comparative conclusions cannot be made, but cfDNA could be a viable prognostic marker [64]. Total cfDNA also showed promise in NSCLC as a prognostic marker [66,67,70]. Total cfDNA concentration may also have utility in monitoring patients after treatment to detect recurrence or disease progression, with total cfDNA elevation being detected earlier than serum biomarkers [50,59,60,65]. 

The consensus for the application of total cfDNA concentration for monitoring is inconclusive with some studies showing decreased cfDNA [38,62,63,65,69,71,72,73] and others showing no correlation/increased cfDNA [67,68,70] while monitoring response to treatment. This discrepancy in cfDNA concentration could be explained by the increase of apoptosis and therefore cfDNA induced by chemotherapy, which would confound the results of these studies [67,68,70]. Another plausible reasoning could be related to different time points in sample collection and different chemotherapies utilized across trials [68]. 

While CTCs and ctDNA are often more accurate measures of tumor burden, cfDNA concentration may still have a place in the management of cancer. Total cfDNA concentration is cost-effective compared to ctDNA or CTCs, which require expensive assays. Due to the repetitive nature of monitoring during and after treatment, this cost is amplified over time [62,63,73]. Because most clinical labs lack CTC platforms, total cfDNA would be easier to implement in clinical settings with greater accessibility [62,73]. Monitoring ctDNA with specific targeted sequences presents a unique challenge as the vast genomic diversity of cancer and additional mutations driven by therapeutic selective pressure, make it difficult to identify effective target sequences [62]. However, measuring cfDNA concentration does not rely on known mutations, and could give a more holistic view of the tumor burden during treatment. In the future, total cfDNA concentration should continue to be studied in applications of prognosis and monitoring with plans to move forward with large-scale studies for clinical validation. However, large-scale clinical validation will require a standardization of the methodology to increase reproducibility in the clinical setting.

## 4. ctDNA

While cfDNA can be increased in healthy patients for various reasons, ctDNA detection is more specific to tumors. It has been established that ctDNA matches sequencing from tumor tissue, with the concordance of mutations in ctDNA and tumor tissue greater than 60–80% in various cancers [45,74,75,76,77,78,79]. Mutations are sequenced from ctDNA through targeted analysis or whole genome sequencing. The ctDNA can also be differentiated from cfDNA through analysis of aberrant methylation.

### 4.1. Techniques

Targeted sequencing looks for specific gene mutations or rearrangements that are common in a particular tumor type, which requires prior knowledge of the region of interest to design the proper assay. Generally, targeted sequencing uses PCR-based methods, such as droplet digital PCR (ddPCR), beads, emulsions, amplification, magnetics (BEAMing), and amplification refractory mutation system (ARMS) qPCR. In ddPCR, sample DNA molecules are separated into droplets, which are amplified by end-point PCR. Then, using fluorescent probes, positive and negative reactions are quantified where the copy number of target DNA is quantified by comparing the two [80]. The analytical sensitivity of ddPCR was 0.001%, detecting 1 mutant copy in 100,000 wild-type copies [80]. With BEAMing, sample DNA is separated onto beads with common primers attached, which are emulsified in oil. The strands attached to primers are then amplified, centrifuged, and collected with a magnet [81]. The beads can then be analyzed for mutations with fluorescent probes and flow cytometry, which can detect target DNA with a sensitivity of 0.01% (1/10,000 copies) [81,82,83]. In ARMS qPCR, target DNA is amplified with primers that are complementary to the mutant sequence. If the primer does not anneal properly then extension does not occur, with qPCR detecting the extension of target DNA [84]. ARMS qPCR can detect target DNA with a sensitivity of 0.1% (1/1000 copies) [85]. Another method using a specialized polymerase, SNPase-ARMS qPCR was able to achieve a sensitivity of 0.001% (1/100,000+ copies) [86]. 

There are also methods that apply next-generation sequencing (NGS), which has higher throughput than PCR, to a target region, such as Targeted Error Correction Sequencing (TEC-Seq), Tagged-Amplicon deep sequencing (TAm-Seq), Safe-Sequencing System (Safe-SeqS), CAncer Personalized Profiling by deep sequencing (CAPP-Seq), or Personalized Analysis of Rearranged Ends (PARE). TEC-Seq uses primers with predetermined barcodes for targeted capture of multiple regions and deep sequencing of captured DNA fragments. TEC-Seq was able to detect 100% and 89% of mutations present at 0.2% and 0.1% mutation allele frequency (MAF) respectively [40]. TAm-Seq uses specialized primers to amplify regions of interest, identifying mutations of 2% MAF with a sensitivity of over 97% [87]. Safe-SeqS adds a unique identifier (UID), which is able to detect target DNA with a sensitivity of 0.05% (1/2000) [88,89] CAPP-Seq combines optimized library preparation with bioinformatics to create a “selector”, which reflects recurrent mutations. The selector is applied to tumor DNA to identify the patient’s mutations, and then applied to ctDNA for quantification, detecting target DNA down to 0.02% (1/5000 copies) MAF [90]. PARE uses mate-paired tags and NGS to find rearranged sequences with a sensitivity of 0.001% (1/100,000+ copies) [91].

DNA from tumors also have aberrant DNA methylation, which occurs early during tumorigenesis, provides information on the tumor’s origin, and are homogenous across populations [27,92,93]. Almost every tumor type is characterized by progressive CpG-island-specific hypermethylation and global CpG hypomethylation [94]. The GRAIL test uses whole genome bisulfite sequencing (WGBS) targeted at over 100,000 methylation regions combined with machine learning to detect cancer and predict tissue of origin (TOO) localization detecting ctDNA down to 0.023% (~1/5000 copies) [95,96]. The OverC test uses an altered WGBS technique called enhanced linear splinter amplification sequencing (ELSA-seq), which mitigates the damage to DNA caused by bisulfite treatment and is able to detect ctDNA down to 0.02% (1/5000) [97,98]. Similarly, the PanSeer assay relies on whole genome bisulfite sequencing with 595 regions sequenced at higher depth, being able to detect cancer DNA with a sensitivity of 0.1% (1/1000 copies) [99]. The higher depth is achieved with newer methodologies, such as cell-free methylated DNA immunoprecipitation and high throughput sequencing (cfMeDIP-seq). cfMeDIP-seq uses beads that bind to methylated DNA and NGS to analyze CpG methylation patterns [100] (Table 2).

The technology used to differentiate ctDNA from cfDNA has continued to develop at a breakneck pace with the “Agilent Resolution ctDx FIRST assay” being approved by the FDA for use as a companion diagnostic tool on 12 December 2022 [105]. Other assays that have pre-market approval by the FDA include Myriad’s BRACAnalysis CDx^®^ (19 December 2014), Epi proColon^®^ (12 April 2016), Roche’s cobas^®^ *EGFR* Mutation Test v2 (1 June 2016), FoundationOne^®^Liquid CDx (26 August 2020), and Guardant360^®^ CDx (7 August 2020) [106,107,108,109,110]. The FDA has also granted the Breakthrough Device designation to multiple ctDNA based assays for a variety of uses in the past few years: CancerSeek Assay (2019), Grail’s Galleri™ test (13 March 2019), Inivata’s RaDaR™ Assay (9 March 2021), Bluestar Genomics’ 5hmC Assay (21 March 2021), Natera’s Signatera Assay (24 March 2021), PredicineCARE cfDNA Assay (20 September 2022), Burning Rock’s OverC Multi-Cancer Detection (MCD) Blood Test (3 January 2023), [111,112,113,114,115,116,117]. While not all modes of utilization of ctDNA in cancer management have been clinically validated, there are hundreds of clinical trials on the use of ctDNA for early detection, diagnosis, prognosis, treatment selection, and monitoring.

### 4.2. Early Detection/Screening

Multiple tests have been developed for the purposes of screening or early detection of cancer in asymptomatic patients. Some tests, such as Epi proColon^®^ and Bluestar Genomics’ 5hmC Assay, are geared towards detecting a single type of cancer, and others, such as CancerSeek, Galleri™, and OverC MCD Assay, are used as multi-cancer early detection tests. Overall, the goal of any screening test is to have high sensitivity in the early stages, specificity to limit false positives and lower costs for patient adherence. With multi-cancer detection, there is also an added requirement of determining the tissue of origin (Table 3).

Currently, the only FDA premarket-approved liquid biopsy test used for screening is the Epi proColon^®^, which can be offered to patients who are unwilling or unable to be screened by other recommended methods. It is performed by detecting methylated *SEPT*9 DNA using real-time PCR and a fluorescent probe [107]. Epi proColon^®^ cites three clinical validation trials that led to the FDA’s premarket approval: a multi-center study of 1544 patients across Germany and the U.S. comparing Epi proColon^®^ to colonoscopies for screening of CRC, which achieved a sensitivity of 68% and a specificity of 80% [118]; a study of 290 patients comparing Epi proColon^®^ with a fecal immunochemical test (FIT) found the Epi proColon^®^ test was statistically non-inferior to FIT. The sensitivity for CRC detection was 73.3% for Epi proColon versus 68.0% for FIT. Specificity was 81.5% for Epi proColon and 97.4% for FIT respectively [119]; and finally, a small two-site randomized controlled trial of 413 patients comparing the adherence to the Epi proColon^®^ blood test, which had significantly higher uptake than FIT (99.5% vs. 88.1%) [128]. Other meta-analyses of 8643 and 2271 patients respectively, evaluating the diagnostic Epi proColon^®^ test agree that it has high diagnostic value for CRC, especially with different algorithms, symptomatic patients, and patients with low compliance [129,130]. While the USPSTF acknowledges that there has been more evidence for the effectiveness of the Epi proColon® test, the USPSTF recommendation does not include serum tests “because of limited available evidence”, and more research is needed to continue evaluating the accuracy and effectiveness of these tests [131,132]. To that end, a search of clinicaltrials.gov for proColon shows 7 completed studies and 2 studies currently recruiting. Only the PERT study pertains to CRC, and it plans to recruit 4500 participants to evaluate the long-term performance of Epi proColon with respect to test accuracy and adherence compared to colonoscopies [133]. 

The Bluestar Genomics’ 5hmC Assay is another test used for single cancer type detection of FDA interest, receiving the Breakthrough Device designation in 2021 [114]. In 2020, Bluestar Genomics compared 64 patients with pancreatic ductal adenocarcinoma (PDAC) to 243 patients without cancer and demonstrated the ability to differentiate and classify the methylation patterns of PDAC [134]. Since then, a pre-print article from 2021 used 89 patients with PDAC and 596 control patients to generate a methylation library, and the predictive model was validated against 79 patients with PDAC, 506 patients with other types of cancer, and 163 patients without cancer. When used with the validation patients, the assay achieved an overall sensitivity of 51.9% and specificity of 100.0%, and for patients with new-onset diabetes, the assay achieved a sensitivity of 55.2% and specificity of 98.4% [120]. With Breakthrough Device designation in hand, Bluestar Genomics has announced two new clinical trials, NODMED and EpiDetect studies, that aim to enroll 6550 and 10,000 patients with newly diagnosed type 2 diabetes respectively. [135,136,137]. 

The CancerSEEK Assay developed by John Hopkins University is unique in that it uses SafeSeqS to detect 1933 distinct mutations in ctDNA and a protein biomarker assay to determine the cancer’s tissue of origin (TOO) [104]. In 2018, they studied the CancerSeek Assay, comparing 1005 patients with previously diagnosed stage 1–3 cancers and 812 healthy control patients. The overall sensitivity of CancerSEEK for the eight cancer types (ovary, liver, stomach, pancreas, esophagus, colorectum, lung, and breast) was 70%, ranging from 98% for ovarian cancer to 33% for breast cancer. The sensitivity varied across stages as well, with 78% for stage 3, 73% for stage 2, and 43% in stage 1. The overall specificity was greater than 99% [104]. By combining the detection of ctDNA and protein biomarkers, CancerSEEK increased sensitivity and specificity and localized the cancer to two TOO in 83% of patients and one TOO in 63% of patients [104]. In the DETECT-A study of 10,006 women with no history of cancer, an early version of CancerSEEK, without enhancements from machine learning, was used to screen for cancer. From 10,000 participants, 490 patients had positive baseline testing, 134 patients were confirmed positive by excluding clonal hematopoiesis of indeterminate potential (CHIP) and confirming the original mutation, 127 patients were evaluated by imaging (PET-CT and others), 64 patients had imaging concerning cancer, and 26 patients were diagnosed with cancer. Overall, the DETECT-A assay achieved a sensitivity of 27.1% and specificity of 98.9% [121]. However, the authors noted that newer generations of CancerSEEK have higher sensitivity and specificity, which would not require a two-step baseline and confirmatory test [104,121]. In January of 2023, the ASCEND study concluded, comparing the CancerSEEK assay against 1000 patients with known or suspected cancer and 2000 control patients. The data collected from the ASCEND study will be used to calibrate the CancerSEEK assay with future clinical trials yet to be announced [138].

GRAIL’s Galleri test was developed, tested, and continues clinical validation in the Circulating Cell-free Genome Atlas Study (CCGA) [139]. Initially, the CCGA compared three sequencing assays, targeted sequencing, whole genome sequencing, and methylation profiling (WGBS), in 2402 participants, which were divided into training and test sets. In the training set of 878 patients with cancer and 580 control patients, methylation sequencing using WGBS achieved the highest sensitivity, ranging from 54–92% in various cancers. In the testing set of 576 patients with cancer and 368 control patients, sensitivity ranged from 36% in breast cancer to 74% in pancreatic cancer, with overall specificity being 98% [122]. Following this, a larger study of 6689 participants was again divided into training and testing sets to build on the WGBS methylation-based assay. The training set included 1531 patients with cancer and 1521 control patients, and the validation set included 1264 patients with cancer and 610 control patients. The overall sensitivity for all stages and cancers in both sets matched closely, being 55.2% and 54.9% in training and validation sets respectively. Sensitivity was lower for cancers in stage 1 being 18% rising to 93% for cancers in stage 4. Specificity for both sets was greater than 99% [96]. Towards further clinical validation, a study of 4077 participants, with 2823 patients with cancer, and 1254 control patients, were tested with the refined Galleri WGBS assay and will continue to follow-up for 5 years. In this large, independent validation set, the Galleri test achieved an overall sensitivity of 51.5% increasing with the stage (stage 1 = 16.8%, stage 2 = 40.4%, stage 3 = 77.0%, and stage 4 = 90.1%). The specificity was greater than 99% [123]. The CCGA clinical trial is ongoing and is set to end in March 2024 [139]. More ongoing observational clinical trials using the Galleri test are the STRIVE study, which has enrolled nearly 100,000 women who are undergoing mammograms with the goal of independently validating Galler’s ability to detect and localize cancers in this population, the SUMMIT study, which is evaluating 13,000 participants with a high risk of cancer from smoking, and the REFLECTION study, which observes Galleri’s effect in clinical settings and patients [95,140,141,142]. There is also progress in interventional studies with the PATHFINDER study screening 4033 asymptomatic patients getting a positive predictive value (PPV) of 45% since the last update in 2021 [143]. This closely matches the estimated PPV of 49% [144]. A continuation of the PATHFINDER study (PATHFINDER-2) plans to enroll 20,000 participants to further evaluate the Galleri test in the general population [145,146].

The latest ctDNA-based screening test to be announced as receiving the FDA Breakthrough Device designation is Burning Rock’s OverC MCD blood test early in 2023 [117]. The OverC assay uses a technique called ELSA-seq, which is a type of WGBS, with increased yield, methylome coverage, and reproducibility [97]. Taking this technique, a prospective multicenter study (PROMISE, NCT04972201) compared and combined cfDNA methylation, cfDNA mutation, and microRNA expression assays in the early detection of 9 cancers. Participants were split into training and test sets with 981 and 492 patients respectively. The methylation model performed the best out of the three, with a sensitivity of 72.4% and a sensitivity of 99.2% overall. The combination of three tests correctly predicted the TOO 75.3% of the time, with 90.9% of cases being in the top two predictions [124,147]. Further development of the ELSA-seq technique occurred in the THUNDER study: a prospective, multi-center study including 1108 participants. A training set of 274 patients with cancer and 195 control patients and a validation set of 351 patients with cancer and 288 control patients were created. In the validation set, the sensitivity of ELSA-seq for patients with early-stage cancer (1–3) was 80.6%, the specificity was greater than 99% and correctly predicted the TOO in 81% of cases [125]. In the single-blind test phase of the THUNDER study, an independent validation set of 360 patients with 202 patients with cancer and 158 control patients used ELSA-seq achieving a sensitivity of 74.8%, specificity of 98.1%, and identified the TOO in 80.8% of cases [126]. An update to the THUNDER study used a training and validation set to create 2 tests with different predetermined cutoffs. Then, the two predictive models were used on an independent validation set of 505 patients with cancer and 505 control patients. Test 1 achieved a sensitivity of 76.2%, a specificity of 98.9%, and predicted the TOO in 79.1% of cases. Test 2 achieved a sensitivity of 70.2%, a specificity of 99.3%, and predicted the TOO in 83% of cases [127]. While the THUNDER study has been completed, Burning Rock plans to perform 2 larger multicenter studies with an estimated 14,000 and 11,879 participants to continue developing and expanding the capabilities of the OverC test to include more types of cancer [148,149]. They are also recruiting for a large prospective, multicenter, interventional study of approximately 12,500 participants, which will evaluate the performance of the OverC MCD blood test in asymptomatic individuals with an increased risk of cancer [150].

Through the FDA approval of Epi proColon^®^’s use in screening for CRC in patients who are unwilling to undergo other recommended screening methods, they confirm the niche that liquid biopsies have in the screening space. However, the reluctance of the FDA and USPSTF to approve and recommend other liquid biopsy screening tests shows there is still a long way to go before overcoming challenges in sensitivity, clinical validation, and cost. One challenge that was previously a concern was the lower specificity of detecting ctDNA due to CHIP, which are mutations associated with myeloid cancers and frequently mutated in healthy patients [151]. However, DETECT-A was able to exclude CHIP mutations and achieve a specificity of 98.9% [121]. Overall, the specificity of ctDNA-based liquid biopsy tests were ~99% with a few exceptions. Most notably, Epi proColon had a specificity of 80% [118] (Table 3).

Of greater concern for the utility of liquid biopsies is the lack of sensitivity, which could range from 27.1% to 80.6%, and gets lower with earlier-stage cancers (Table 3). Low ctDNA shedding tumors with low tumor burden or less metastatic spread are likely contributors to lowering the sensitivity of the test through sampling bias [152]. While various tests have been analytically validated to detect ctDNA in MAF down to 1/100,000, clinically, once the MAF drops below 0.01% (1/10,000 copies) the use of a 10 mL sample of blood will not contain a single ctDNA fragment to sequence or detect [153]. However, newer approaches are working on increasing the sensitivity of the liquid biopsy tests. Multi-omics combines the analysis of ctDNA with other biomarkers, such as fragmentation length or serum proteins. Based on a shorter fragment length ctDNA, Mouliere et al. were able to increase the sensitivity to 90% without a loss in specificity of 98%. However, there was sometimes a loss of sequencing data from ctDNA because it was not integral to detecting the tumor [32]. Similarly, a machine learning approach was able to achieve a sensitivity of 95.5% and specificity of 95% in 971 early-stage cancer patients [154]. CancerSeek used multi-omics as described above and reported an increase in sensitivity and specificity after applying serum proteins to the test [104]. Another study found similar results in pancreatic cancer increasing the sensitivity from 30% to 64% [154]. 

Another prospect to increase sensitivity might be to increase the sample volume. It is estimated a blood sample of 150 to 300 mL would be required to achieve a sensitivity of 95% for screening early breast cancer, which is not possible with a simple blood draw [155]. A method already being studied in the collection of CTCs is the application of an apheresis machine to collect a larger quantity of cells than would be possible through simple blood drawing. In advanced prostate cancer, Lambros et al found an increase in CTC yield from 167 per patient to 12,546 [156]. In a similar study on breast cancer, they found a 205× increase in the yield of CTCs [157]. For the application of apheresis in ctDNA, researchers from Johns Hopkins University have developed a system that can capture ctDNA from the flowing unaltered plasma running through an apheresis machine [158]. However, further development of the technology is needed [158].

Despite the continuous improvement in the analytical and clinical ability of liquid biopsies, none of the tests above have been recommended by the USPSTF. In part, this is due to the limitations of screening for some types of cancer, but the USPSTF cites a lack of clinical validation in its recommendation for epi proColon [131,159]. With that being said, there are over 200,000 patients being recruited across 10 clinical trials, including interventional trials used to determine the actual clinical benefits of the tests.

### 4.3. Treatment Selection/Companion Diagnostics

The FDA defines a companion diagnostic device as an in vitro test that provides information that is essential for the safe and effective use of a corresponding therapeutic product [160]. These are tested analytically and clinically to ensure that companion diagnostic devices do not lead to withholding appropriate therapy or administering inappropriate therapy [160]. In an era of increasing precision medicine, companion diagnostics are an integral part of identifying targetable mutations in the patient’s cancer. Recognizing this, the FDA has given 5 ctDNA-based tests pre-market approval. Some are PCR-based tests, such as Roche’s cobas^®^ *EGFR* Mutation Test v2, and Qiagen’s therascreen test. Some are NGS-based tests, such as the FoundationOne^®^Liquid CDx, Guardant360^®^ CDx, and Agilent’s Resolution ctDx FIRST assay. The PredicineCARE cfDNA Assay has received the FDA Breakthrough Device designation and also utilizes NGS (Table 4).

#### 4.3.1. *BRCA*

While not detecting ctDNA, the first liquid biopsy-based companion diagnostic test was Myriad’s BRACAnalysis CDx^®^, which is approved for the identification of deleterious germline *BRCA* (*gBRCA*) variants from whole blood products in breast, ovarian, pancreatic, and metastatic castration-resistant prostate cancer (mCRPC), using Sanger sequencing and multiplex PCR. The results provide aid in identifying eligible patients for a PARP inhibitor, such as olaparib, talazoparib, or rucaparib [106]. The initial FDA priority review hinged on the completion of a randomized, double-blind, placebo-controlled, phase 2 study of olaparib, which showed a significantly prolonged progression-free survival with *BRCA* mutation-positive patients most likely to benefit from treatment [179]. In a pooled analysis of 300 patients from phase 1 and 2 trials, olaparib elicited durable responses in patients with relapsed *gBRCA* mutations [180]. The FDA analysis noted a limited representation of *BRCA1/2* variants in the previous clinical trials and requested data from the ongoing clinical trials [161]. In those follow-ups, confirmatory, phase 3 trials (SOLO2, SOLO3), olaparib demonstrated a clinically significant survival benefit in the same patient population [181,182]. In the initial approval of BRACAnalysis, Myriad provided evidence of the analytical sensitivity and specificity being ~99% for both, and the clinical validity of BRACAnalysis was confirmed through a concordance rate between local test results and BRACAnalysis of 96.7% during clinical trials [161]. In the subsequent extension of BRACAnalysis’s indications to other PARP inhibitors and cancer types, some of the clinical trials used BRACAnalysis as part of the inclusion criteria, and that showed the efficacy of drugs being tested on the specific type of cancer [162,163,164]. In the other indications, the clinical trials cited use Foundation Medicine’s NGS-based assay. The study of Rucaparib in ovarian carcinoma (ARIEL2) used the Foundation Medicine T5 NGS assay, and the study of olaparib in mCRPC (PROfound) used the FoundationOne CDx NGS assay. Both of the aforementioned tests use solid tissue biopsy samples and included mutations other than *BRCA1/2* [165,166,167,168]. While niraparib was not officially on the list of companion diagnostic indications, the final label update for BRACAnalysis notes that in a randomized, double-blind, phase 3 trial testing niraparib on 203 patients with g*BRCA* mutations, and 350 patients without g*BRCA* mutations the progression-free survival for the g*BRCA*+ cohort was significantly longer than the non-g*BRCA* cohort. This study used the myChoice HRD test on solid biopsy samples to identify the *BRCA1/2* mutations [183,184].

NGS from Foundation Medicine was approved by the FDA on 19 December 2016 in the form of the FoundationFocus CDx*BRCA* Assay for use in detecting *BRCA1/2* alterations following the Rucaparib ARIEL1/2 clinical trials [185]. The initial study enrolled patients with *BRCA* mutations detected by local testing. Later, the *BRCA* mutations were confirmed at a central Foundation Medicine lab using the Foundation Medicine T5a Panel [167,185]. Similar to BRACAnalaysis, the approval of FoundationOne Liquid CDx was based on a bridging study measuring the concordance compared to the clinical trial tissue assay, as well as the effectiveness of the FoundationOne liquid biopsy test to select patients for treatment with rucaparib [169,174]. When 217/491 patients from the ARIEL2 patient population were tested, the positive percent agreement was 93.8% (60/64) and the negative percent agreement was 97.4% (149/153) [169]. In the patients treated with rucaparib, there were 26 *BRCA*-positive patients as determined by FoundationOne^®^ Liquid CDx and 61 *BRCA*-positive patients as determined by the clinical trial assay. The overall response rate was similar in the two groups being 53.8% (14/26) and 54.1% (33) respectively [169]. The bridging studies were conducted in accordance with the approval of the FoundationOne Liquid CDx assay for use in identifying *BRCA*-positive mCRPC for treatment with rucaparib [170,186]. The FoundationOne Liquid CDx assay is also approved for use in identifying *BRCA*-positive mCRPC, but no bridging study was cited by the FDA in the summary sheet [170].

#### 4.3.2. *EGFR*

Similar to complementary diagnostic tests for detecting *BRCA* mutations, tests for detecting *EGFR* mutations required both analytical and clinical validation. Roche’s cobas^®^ *EGFR* Mutation Test v1 was an RT-PCR test for the detection of exon 19 deletions or exon 21 L858R missense mutations in non-small cell lung cancer (NSCLC), for which erlotinib is indicated [187]. The EURTAC study was a phase 3 trial that screened 1044 patients using a clinical trial assay with a combination of methods for comparing erlotinib vs. cisplatin chemotherapy [188]. The FDA then approved the tissue-based cobas^®^ *EGFR* Mutation Test v1 test based on retrospective testing of 487 samples, which had 432 results that could be compared to the clinical trial assay [187,189]. When compared, the cobas^®^ *EGFR* Mutation Test v1 had an overall percent agreement of 96.3% (416/432) [189]. While retrospective analysis of the progression-free survival from the EURTAC study was not completed, the cobas^®^ *EGFR* Mutation Test v1 was used to enroll 217 patients for treatment with erlotinib vs. gemcitabine/cisplatin showing a significant increase in PFS using erlotinib in *EGFR* mutation-positive patients [190]. The ASPIRATION and FAST-ACT2 studies also supported the effectiveness of erlotinib in patients with *EGFR* mutations detected by the cobas^®^ *EGFR* Mutation Test v1 [191,192]. In another bridging study, Roche retrospectively tested samples from the ENSURE, ASPIRATION, and FAST-ACT2 studies with the cobas^®^ *EGFR* Plasma Test v2 [101,171]. In a pooled analysis of the 897 paired samples available, an imperfect concordance with a positive predictive agreement (PPA) of 72.1% (339/470) and negative predictive agreement (NPA) of 97.9 (418/427) was recorded [171]. However, a negative plasma test would lead to patients using a solid tissue biopsy to determine the *EGFR* status, so a positive predictive value (PPV) of 97.6% was enough to approve the first liquid biopsy test for detecting exon 19 deletion and exon 21 L858R mutation in 2016 - the cobas^®^ *EGFR* Plasma Test v2. [193]. With osimertinib, cobas *EGFR* Mutation Test v1 was used in the inclusion criteria for the AURA2 phase 2 clinical trial [194]. Then, in a bridging study, the cobas *EGFR* Plasma Test v2 test detected a T790M gatekeeper mutation with a PPA of 56.8% and a NPA of 80.2% [172]. Osimertinib was also established as a first-line treatment option for patients with exon 19 deletion and L858R mutation, where both plasma and tissue are now used [195,196,197]. Plasma or tissue was also approved as a companion diagnostic for gefitinib, but no trials are cited in the approval [173].

In a deviation from previous approvals that paired a single test result with a single drug based on clinical trials and bridging studies, the FDA approved companion diagnostic tests that detected the exon 19 del or L858R mutation in *EGFR* as suitable for treatment with a tyrosine kinase inhibitor (TKI) approved for that indication [198]. On 27 October 2020, the new group labeling for the cobas^®^ *EGFR* Mutation Test v2 added two new tyrosine kinase inhibitors to its indication when detecting exon 19 deletion or L858R mutation with afatinib or dacomitinib. However, these would not be added to the plasma test [198]. In a similar progression, the FoundationOne CDx test was approved based on a bridging study comparing concordance with the cobas^®^ v2 *EGFR* mutation test for the detection of *EGFR* exon 19 deletions, L858R, and T790M substitutions being eligible for treatment with afatinib, gefitinib, erlotinib, or osimertinib [199,200]. Another bridging study was later completed to approve the Foundation One Liquid CDx, with a PPA and NPA of above 95% across multiple tests, for the detection of *EGFR* exon 19 del, and L858R mutation with gefitinib, osimertinib, and erlotinib [169,185]. Again, the group approval in 2022 of FoundationOne CDx added tyrosine kinase inhibitors, dacomitinib, to its indication, but did not add new TKIs, afatinib or dacomitinib, to FoundationOne Liquid CDx’s [201,202]. Another liquid biopsy-based test, Guardant360 liquid biopsy CDx, was approved for the detection of *EGFR* alterations for use in only one TKI, osimertinib. The approval of Guardant360 CDx came from a bridging study measuring the concordance between Guardant360 and the cobas^®^ *EGFR* mutation test using samples from the FLAURA clinical trial with a PPA of ~75% and NPA of ~99% [175]. 

#### 4.3.3. Other

The Guardant360 CDx has also been approved for the detection of *KRAS* G12C mutations to indicate usage of sotorasib based on concordance studies comparing samples from the CodeBreaK100 study originally analyzed with the Therascreen *KRAS* RGQ PCR Kit [176,203,204]. With 189 patients compared, the PPA was 70.7% (82/116) and the NPA was 100% (73/73). Importantly, the Guardant360 CDx had no false positives and an overall response rate in the positive patient population of 38% [176,177]. The newest approval from the FDA of the Agilent Resolution ctDx FIRST assay was also for the detection of the *KRAS* G12C mutation for use with adagrasib, another *KRAS* inhibitor [178]. In a bridging study measuring the concordance between the Resolution FIRST assay and locally confirmed detections from the KRYSTAL-1 study, they compared 223 samples getting a PPA of 87% (47/54) and NPA of 97.6% (165/169) [178,205,206,207]. 

Predicine announced that the FDA granted the breakthrough device designation to the PredicineCARE companion diagnostic assay on 20 September 2022 [116]. This followed a clinical trial using their NGS-based assay to detect mutations in ctDNA in different types of breast cancer [208]. Among 141 patients with advanced breast cancer, 112 (79.4%) had plasma samples with mutations detected. Then, 21 patients had solid biopsies to compare to their plasma samples with the same single nucleotide variant detected in 6 plasma samples out of 10 tissue samples (6/10) in the *PIK3CA* gene, 5/9 in *TP53*, and 5/6 in *ERBB2* [209]. While a study in 2020 showed strong concordance between the PredicinePLUS NGS assay and Guardant360, it only included 15 patients and should be carried out with a set PredicineCARE assay as they move forward [210].

With the ongoing introduction and approval of numerous companion diagnostic tests and the expansion of clinical utility through grouping therapeutic products to determine the change in labeling for companion diagnostic devices from specific products to a group of therapeutic products, the FDA added guidance for the industry in April 2020 [211]. At the heart of the guidance is still ensuring proper labeling to ensure the correct identification of patients for therapeutic treatment, which is now a specific group of products approved for the same indications based on molecular alterations. The goal of the FDA’s newer grouped approval comes with the following considerations: identifying the specific alterations that include or exclude the use of a group of therapeutics, identifying at least two approved therapeutics for those alterations, and demonstrating analytical and clinical validation [211]. Having the grouped labeling of both companion diagnostic devices and therapeutic treatments should make it easier for both products to gain traction and utility. Therapeutic treatments that are linked to a specific molecular alteration could be detected by multiple companion diagnostic devices. A new companion diagnostic device that can detect molecular alteration could be used for indicating multiple treatments. For instance, Guardant360 is already approved for use with one TKI and could make the argument to expand its coverage to all TKIs.

### 4.4. Minimal Residual Disease

In patients with curative disease, monitoring after treatment generally consists of serial measurements of various serum proteins and serial radiographic imaging [212,213]. However, there are some gaps in the utility of certain serum proteins, such as in pancreatic and breast cancer, as previously discussed [52,53,57]. Serum proteins also run into issues with limited sensitivity and specificity [212]. While imaging improves the detection of recurrence, it can only detect macroscopic disease [214]. Also, imaging is associated with a risk of radiation exposure and inconclusive readings due to normal reactions to treatment [212,215]. Analysis of ctDNA has shown utility for detecting relapse and resistance mutations before clinical progression occurs [216,217,218]. The FDA has given the breakthrough device designation to two ctDNA-based tests: Inivata’s RaDaR™ Assay and Natera’s Signatera Assay (Table 5).

Inivata’s RaDaR™ Assay uses an enhanced version of the TAm-Seq technology, which originally covered 6 genes with a 97% sensitivity and specificity at 2% MAF [87]. The InVisionFirst assay used 36 genes targeted for NSCLC and was able to detect target DNA down to 0.25 MAF with a sensitivity of 92.46% [226]. Building on that, the RaDaR Assay uses personalized targets based on whole genome sequencing of tumor tissue, which amplifies up to 48 patient-specific tumor variants from cfDNA [102]. With 48 variants, the assay had a sensitivity of 100% at 0.002% MAF, and with only 16 variants, the assay had a sensitivity of 95% at 0.004% MAF [102]. This approach has since been studied for the detection of MRD in NSCLC, head and neck squamous cell carcinoma (HNSCC), breast cancer, and melanoma. In the “LUng cancer—Circulating tumor DNA” (LUCID) study, 88 patients had their ctDNA levels measured before and after treatment with surgery [227]. For the 77 patients undergoing observation >2 weeks after the end of treatment, ctDNA was detected with a clinical specificity of 98.7% (150/152) samples in 64.3% (18/28) patients who later experienced a clinical recurrence of their first tumor, with lead times of ~200 days from detection to disease recurrence [219]. For HNSCC, 17 patients with stage 3 or 4 diseases had samples taken before and after surgery with chemoradiotherapy as needed [220]. With later-stage diseases, the RaDaR assay was able to detect pre-operative ctDNA in 100% of patients, and ctDNA was detected 108 to 253 days before clinical recurrence occurred in 100% (5/5) of patients with no false positives [220]. In a larger study of 38 patients with recurrent or metastatic HNSCC, the RaDaR assay was used to monitor patients on platinum-based chemotherapy or immunotherapy [228]. Preliminary data from this study has shown that a decrease in ctDNA MAF from baseline to after first treatment was correlated with improved PFS [229]. Out of 22 early-stage breast cancer patients undergoing surgical treatment, RaDaR successfully identified MRD in 100% (17/17) of patients with clinical recurrence and no false positives [221]. In a follow-up study of 38 patients, the RaDaR assay detected 92% (12/13) of distant recurrence and 38% (3/8) of local recurrence. However, in the 6 cases missed by the RaDaR assay, 100% had indications of a possible alternative origin or second primary tumor. Only 1 patient of 17 who did not have disease recurrence had detectable ctDNA at 0.0085% MAF. [222]. There are also two clinical trials with ongoing monitoring. The TRACER trial, which is a sub-trial of the LIBERATE trial, has recruited 145 patients with breast cancer with the percentage of patients with detectable ctDNA dropping with treatment of various neoadjuvant therapy after surgery [230,231]. The other trial, cTRAK-TN, has developed RaDaR assays for 142 patients and has achieved a longer median lead time and higher sensitivity compared to dPCR [232]. In another ongoing trial of 47 patients with melanoma, 12 patients had ctDNA detected after surgical treatment with detection possibly preceding recurrence and being associated with lower PFS and OS [233]. While not monitoring for post-treatment MRD, RaDaR is being studied in advanced urothelial cancer for monitoring during treatment with immunotherapy with decreases in plasma ctDNA levels being associated with a pathological response [234].

Natera’s Signatera Assay is also based on a personalized targeted multiplex PCR-based NGS technology, which can track up to 16 clonal variants from cfDNA in plasma [103,223]. In the initial studies applying the technology to 49 patients with breast cancer, the Signatera Assay was able to detect ctDNA down to MAF of 0.01% and ctDNA was detected in 94% (16/17) patients 0.5–24 months prior to distant metastatic recurrence with detection in 89% (16/18) of overall recurrence. Again, there were no false positives from this study [103]. Similar findings were demonstrated in 122 early-stage CRC and 68 urothelial bladder carcinoma patients detecting 87.5% (14/16) of recurrence with an average lead time of 8.7 months and 100% (13/13) of recurrence with a median lead time of 96 days respectively [223,224]. In the urothelial bladder carcinoma study, the specificity was 98% (48/49) [224]. In another study of 112 patients with mCRC undergoing metastatic resection, 96.7% (59/61) of patients with detectable ctDNA had disease progression with a median lead time of 3.16 months. With one sample collected, the sensitivity was lower at 72% (59/82) and specificity of 93.3% (28/30). With two samples the sensitivity was up to 91.4% (32/35) and no change in specificity of 93.3% (14/15) [225]. In CIRCULATE-Japan, their biggest study of CRC, Signatera was used in 1039 patients with stage 2–4 resectable CRC to determine eligibility for adjuvant therapy after surgery based on the risk of recurrence [235,236]. In the cohort of 1039 patients, ctDNA positivity 4 weeks after surgery was associated with much higher rates of recurrence than ctDNA negative patients, 61.4% (115/187) versus 9.5% (81/852) respectively [237]. They also found that ctDNA positivity was better than CEA for relapse detection [237]. Monitoring for MRD is still ongoing for the CIRCULATE-Japan study, and another application of the Signatera test, the BESPOKE CRC study, is a multicentre study in California that plans to recruit 2000 patients to determine adjuvant therapy [238,239]. In a follow-up in urothelial carcinoma, 581 patients were monitored for MRD to determine adjuvant therapy that detected ctDNA prior to radiological relapse 59% of the time [240]. 

Similar to the use of ctDNA-based liquid biopsies in screening, the detection of ctDNA is highly specific for the presence of cancer, with the detection of MRD having a specificity >90% in all the completed studies [103,219,220,221,222,224,225]. (Table 5) The decrease in specificity is likely due to the repetition of the test over a shorter time frame, and possible detection of dormant disease [222]. With the detection of MRD, clinicians can obtain information on disease processes weeks and months ahead of current monitoring by serum proteins or imaging [225,231]. For example, the MRD tests have promising utility in monitoring for response to systemic therapy [224,234,237,240]. However, with the high specificity comes a lower sensitivity. The analytical sensitivity of both the RaDaR Assay and Signatera Assay are impressive being 0.002% MAF and 0.01% MAF respectively [102,222]. However, the clinical sensitivity could range from 64.3% (18/28) to 100% (13/13) [219,221,224]. (Table 5) The sensitivity of the tests increased with the disease burden being better at detecting distant over local recurrence [103,222]. Sensitivity also increased with more longitudinal samples with little decrease in specificity [225]. The increase and decrease in sensitivity of these tests fall in line with our understanding of sampling bias as discussed earlier [152,153]. Increasing the sample size by taking samples every month should increase the likelihood of detecting a rare ctDNA fragment when disease burden is low, such as in MRD. The future application of ctDNA in the detection of MRD is likely unhampered by the sensitivity, specificity, or lead times, especially as the technology continues to be refined. However, the most important part of getting the technology into the clinic will be the clinical trials that are ongoing, and future expansions for clinical validation [230,231,235,236,238,239].

## 5. Conclusions

The clinical applications of quantitative and qualitative analysis of cfDNA using liquid biopsy can be exciting solutions to weaknesses of solid biopsy, serum proteins, and imaging. While the clinical utility of measuring total cfDNA as a screening tool is low, there are promising studies using it to determine patients’ prognosis, treatment response, and recurrence. However, total cfDNA being increased in normal physiological processes, a lack of consensus for levels of cfDNA during treatment, and an overall lack of standardization in methodology holds it back from being studied in large-scale clinical trials. In contrast, a variety of ctDNA-based companion diagnostic devices have been clinically validated and are used in clinics under pre-market approval from the FDA for treatment selection. An increase in positive percent agreement might increase the clinical utility of these companion diagnostic devices, but the innovation has mostly been in the addition of group approvals, which will make it easier for the industry to create and push new companion diagnostic devices to market. Using ctDNA detection for other parts of clinical management, such as screening and MRD, is promising with clinical trials completed showing high specificity for detecting cancer. However, several factors, such as low tumor burden and sampling bias, contribute to low sensitivity, but there are new innovations in technology, multi-omics, and sampling to find solutions to these issues. With that said, larger clinical trials using ctDNA-based liquid biopsy tests are still ongoing to prove their clinical validity and propel them to mainstream use.

## Figures and Tables

**Figure 1 ijms-24-13219-f001:**
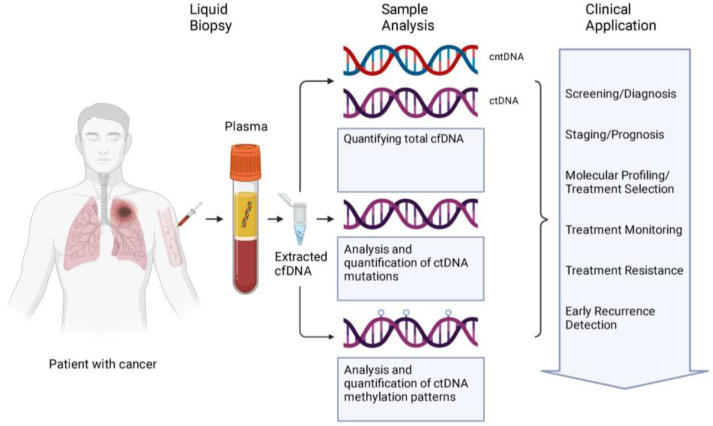
The cfDNA from liquid biopsy samples are analyzed in several ways: quantification of total cfDNA, qualitative and quantitative analysis of ctDNA mutations, and quantitative analysis of ctDNA methylation patterns. The analysis is then applied in the clinic at every step of patient management from screening to recurrence. Abbreviations: cntDNA: circulating non-tumor DNA; ctDNA: circulating tumor DNA; cfDNA: cell free DNA. Created with Biorender.com (accessed on 10 August 2023).

**Table 1 ijms-24-13219-t001:** Studies using total cfDNA to predict patient prognosis.

Cancer Type	# of Patients	cfDNA concentration	PFS HR	OS HR	Additional Info	Citation
PDAC	74	>9.71 ng/mL	6.85	4.16	Average of cfDNA concentration before and after chemotherapy treatment.	[59]
PDAC	-	-	1.96	3.39	Subgroup analysis of cfDNA during treatment.	[60]
PCa	-	>cut-off	log(HR) = 0.84	log(HR) = 0.60	Meta-analysis of studies using different cut-off points.	[61]
CRPC	-	>cut-off	log(HR) = 0.65	log(HR) = 0.59	Meta-analysis of studies using different cut-off points.	[61]
MBC	194	>0.306 ng/uL	1.193	1.199	Majority of samples were collected during treatment.	[62]
MBC	117	high cfDNA	1.64	2.73	High cfDNA was determined by comparing to previous samples. HR based on comparing low and high cfDNA levels.	[63]
mCRC	1076	>cut-off	-	2.39	Meta-analysis of studies using different cut-off points.	[64]
mCRC	43	>26 ng/mL	1.51	2.02	PFS and OS determined from samples before treatment.	[65]
NSCLC	-	>cut-off	1.32	1.64	meta-analysis of studies using different cut-off points.	[66]
NSCLC	177	>70 ng/mL	2.6	2.63	PFS and OS determined from samples before treatment.	[67]

Abbreviations: PDAC: pancreatic ductal adenocarcinoma; PCa: prostate cancer; CRPC: castration resistant prostate cancer; MBC: metastatic breast cancer; mCRC: metastatic colorectal cancer; NSCLC: non-small cell lung cancer.

**Table 2 ijms-24-13219-t002:** The analytical sensitivity of different technologies used to differentiate ctDNA from cfDNA.

Technology Used	Analytical Sensitivity	Cancer Type	Test	Citation
RT-PCR	0.1–1% MAF	NSCLC	cobas *EGFR*	[101]
ddPCR	0.001% MAF	BRAF V600E		[80]
BEAMing	0.01% MAF	CRC, MBC		[82,83]
ARMS-qPCR	0.1% MAF	mCRC		[85]
SNPase-ARMS qPCR	0.001% MAF	Melanoma		[86]
TEC-Seq	0.1% MAF	BC, LC, OVC, CRC		[40]
TAm-Seq	0.002% MAF	Various	RaDaR	[102]
TAm-Seq	0.01% MAF	BC	Signatera	[103]
SafeSEQ	0.05% MAF	Various	CancerSEEK	[88,89,104]
CAPP-Seq	0.02% MAF	NSCLC		[90]
PARE	0.001% MAF	BC, CRC		[91]
WGBS	0.023% MAF	Various	Galleri	[95,96]
ELSA-seq	0.02% MAF	Various	OverC	[97,98]
cfMeDIP-seq	0.1% MAF	Various	PanSeer	[99]

NSCLC: non-small cell lung cancer; CRC: colorectal cancer; MBC: metastatic breast cancer; mCRC: metastatic colorectal cancer; BC: breast cancer; LC: lung cancer; OVC: ovarian cancer; MAF: mutant allele frequency; RT-PCR: real-time PCR; ddPCR: droplet digital PCR; BEAMing: beads, emulsions, amplification, magnetics; ARMS-qPCR: amplification refractory mutation system qPCR; TEC-Seq: targeted error correction sequencing; TAm-Seq: tagged-amplicon deep sequencing; Safe-SeqS: safe-sequencing system; CAPP-Seq: cancer personalized profiling by deep sequencing; PARE: personalized analysis of rearranged ends; WGBS: whole genome bisulfite sequencing; ELSA-seq: enhanced linear splinter amplification sequencing; cfMeDIP-seq: cell-free methylated DNA immunoprecipitation and high throughput sequencing.

**Table 3 ijms-24-13219-t003:** Sensitivity and specificity of ctDNA-based screening tests in various cancer types.

Test	# of Patients *	Cancer Type (Stage)	Sensitivity	Specificity	Citation
Epi proColon	1544	CRC	68%	80%	[118]
Epi proColon	290	CRC	73.3%	81.5%	[119]
Bluestar	748	PDAC	51.9%	100%	[120]
CancerSEEK **	9911	Various	27.1%	98.9%	[121]
CancerSEEK	1817	Various (1–3)	70%	>99%	[104]
Galleri	944	Various	36–74%	98%	[122]
Galleri	1264	Various	54.9%	>99%	[95]
Galleri	4077	Various	51.5%	>99%	[123]
OverC	492	Various	72.4%	99.2%	[124]
OverC	639	Various (1–3)	80.6%	>99%	[125]
OverC	360	Various	74.8%	98.1%	[126]
OverC	1010	Various (1–3)	68.5%	96.3%	[127]

* Total number of patients in the testing/validation set used to determine sensitivity and specificity. ** An earlier version of the CancerSEEK test was used in this clinical trial. Abbreviations: CRC: colorectal cancer; PDAC: pancreatic ductal adenocarcinoma.

**Table 4 ijms-24-13219-t004:** Companion diagnostic tests and the related concordance testing cited in the FDA Premarket Approval summaries.

Diagnostic Test	Biomarker/ Details	Cancer Type	Drug Name	Clinical Trial #	Compared Test	Concordance	Citations
BRACAnalysis	*gBRCA1/2*	OC	Olaparib	NCT00753545 NCT01874353	local testing	99.2% (259/261)	[161]
BRACAnalysis	*gBRCA1/2*	BC	Olaparib	NCT02000622	BRACAnalysis	n/a	[162]
BRACAnalysis	*gBRCA1/2*	BC	Talazoparib	NCT01945775	BRACAnalysis	n/a	[163]
BRACAnalysis	*gBRCA1/2*	PDAC	Olaparib	NCT02184195	BRACAnalysis	n/a	[164]
BRACAnalysis	*gBRCA1/2*	OC	Rucaparib	NCT01891344	Foundation Medicine T5	n/a	[165,166]
BRACAnalysis	*gBRCA1/2*	PC	Olaparib	NCT02987543	FoundationOne CDx	n/a	[167,168]
FoundationOne Liquid CDx	*BRCA1/2*	OC	Rucaparib	NCT01891344	Foundation Medicine T5	96.3% (209/217)	[169]
FoundationOne Liquid CDx	*BRCA1/2*	mCRPC	Rucaparib	NCT02952534	FoundationOne LDT FoundationOne Liquid LDT local testing	89.4% (144/161)	[170]
FoundationOne Liquid CDx	*BRCA1/2*	mCRPC	Olaparib	NCT02987543	FoundationOne CDx	n/a	[170]
cobas *EGFR* Plasma Test v2	*EGFR* *	NSCLC	Erlotinib	NCT01310036	cobas *EGFR* Mutation Test v1	84.4% (757/897)	[171]
cobas *EGFR* Plasma Test v2	*EGFR* **	NSCLC	Osimertinib	NCT02094261	cobas *EGFR* Mutation Test v1	65.0% (206/317)	[172]
cobas *EGFR* Plasma Test v2	*EGFR* *	NSCLC	Gefitinib	-	-	-	[173]
FoundationOne Liquid CDx	*EGFR* * *EGFR* **	NSCLC	Erlotinib Osimertinib Gefitinib	-	cobas *EGFR* Mutation Test v2	94.4% (167/177)	[174]
Guardant360 CDx	*EGFR* * *EGFR* **	NSCLC	Osimertinib	NCT02296125	cobas *EGFR* Mutation Test v2	79.5% (372/468)	[175]
Guardant360 CDx	*KRAS* G12C	NSCLC	Sotorasib	NCT03600883	therascreen *KRAS* RGQ PCR Kit	82.0% (155/189)	[176,177]
Resolution ctDx	*KRAS* G12C	NSCLC	Adagrasib	NCT03785249	local testing	95.1% (212/223)	[178]

*gBRCA1/2* = germline *BRCA* variants detected from DNA extracted from non-tumor cells. *EGFR* * = *EGFR* exon 19 deletions and exon 21 L858R; *EGFR* ** = *EGFR* T790M. Abbreviations: OC: ovarian cancer; BC: breast cancer; PDAC: pancreatic ductal adenocarcinoma; PC: prostate cancer; mCRPC: metastatic castration resistant prostate cancer; NSCLC: non-small cell lung cancer.

**Table 5 ijms-24-13219-t005:** Clinical trials using minimal residual disease tests to predict recurrence with median lead time.

MRD Test	Cancer Type	# of Patients	Specificity	Sensitivity	Lead Time	Additional Info	Citation
RaDaR™	NSCLC	77	95.9% (47/49)	64.8% (18/28)	212.5 days	ctDNA detected >2 weeks after surgery	[219]
RaDaR™	HNSCC	17	100% (12/12)	100% (5/5)	122 days	ctDNA detected from samples taken during routine follow-up visits	[220]
RaDaR™	BC	22	100% (5/5)	100% (17/17)	386.7 days	ctDNA detected from samples taken during routine follow-up visits	[221]
RaDaR™	BC	38	94.1% (16/17)	71.4% (15/21)	92 days	ctDNA detected from samples taken at detection of recurrence or 3 years later	[222]
Signatera	BC	49	100% (31/31)	89% (16/18)	267 days	ctDNA detected from samples taken every 6 months	[103]
Signatera	CRC	75	98.3% (58/59)	87.5% (14/16)	261 days *	ctDNA detected from samples taken at 1 month and every 3 months.	[223]
Signatera	UC	68	98% (48/49)	100% (13/13)	96 days	ctDNA detected from samples taken during routine follow-up visits	[224]
Signatera	mCRC	112	93.3% (14/15)	91.4% (32/35)	95 days	ctDNA detection with samples from two-time points	[225]

* Only average lead time was reported. Abbreviations: NSCLC: non-small cell lung cancer; HNSCC: head and neck squamous cell carcinoma; BC: breast cancer; CRC: colorectal cancer; UC: urothelial carcinoma; mCRC: metastatic colorectal cancer.

## Data Availability

Data sharing not applicable.

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
