# Peer review of "Using cfDNA and ctDNA as Oncologic Markers: A Path to Clinical Validation"

_ijms, 2023, doi:10.3390/ijms241713219_

Round 1
Reviewer 1 Report
The authors of the manuscript titled “cfDNA and ctDNA as Oncologic Markers: Path to Clinical Validation” studied the utility of measuring total cfDNA, techniques used to differentiate ctDNA from cfDNA, and the utility of different ctDNA-based liquid biopsy kits.
I recommended publication after minor corrections.
The abstract and conclusion need to be re-write. The abstract needs to include more information about what the authors performed and the knowledge that is collected.
The conclusion is short and needs to include more information about what the authors found, the future studies of measuring cf DNA and Ct DNA, and any limitations.
Author Response
Hello,
Thank you for your kind remarks. I have taken note that you wanted to strengthen the abstract and conclusion. I have gone ahead and reworked those with more details on the findings throughout the review. This is mostly a summarization of the end of every section, but I agree that it would be good to put it into one place at the end. Thank you again.
Very Respectfully,
Jonathan Dao

Reviewer 2 Report
I evaluated the whole manuscript and found it a complete and informative article. The manuscript: cfDNA and ctDNA as Oncologic Markers: 2 Path to Clinical Validation by Dao et al. is sound and integrated containing beneficial data. The introduction properly focuses on the significance of liquid biopsy in cancer. Then, the principles, differences, methods for detection, and application of cfDNA and ctDNA are well-explained. The tables are organized, and informative. Information in both the diagnosis and therapy sections is well-represented to give a comprehensive understanding of clinical applications. The study is dynamic and well-written with a good quality of writing. The number of references also indicates how much data is collected for the study. The conclusion may be improved to represent a comprehensive look at the study findings. I found no requirement for any revision or amendments. So, I can endorse its publication.
Author Response
Hello,
Thank you for your kind remarks. I have written the conclusion to get a better look at each section rather than an overall conclusion. Another reviewer also commented to rewrite the abstract, so I rewrote it to include some conclusions as well. Thank you again.
Very Respectfully,
Jonathan Dao

Reviewer 3 Report
The authors deliver a comprehensive overview of the application of cfDNA and ctDNA in liquid biopsy for cancer detection and monitoring. The review is skillfully composed, staying current with recent advancements in the field. Notably, it encompasses the latest developments in liquid biopsy test kits, featuring multiple FDA pre-market approved offerings from various companies. The manuscript seamlessly aligns with the thematic focus of the special issue.
Author Response
Hello,
Thank you for your kind remarks. At the recommendation of the reviewers, I have rewritten the abstract and the conclusion to include more information from each section of the review. Thank you again.
Very Respectfully,
Jonathan Dao
